# Effects of Irrigation and Fertilization on the Morphophysiological Traits of *Populus sibirica* Hort. Ex Tausch and *Ulmus pumila* L. in the Semiarid Steppe Region of Mongolia

**DOI:** 10.3390/plants10112407

**Published:** 2021-11-08

**Authors:** Ser-Oddamba Byambadorj, Byung Bae Park, Jonathan O. Hernandez, Enkhchimeg Tsedensodnom, Otgonsaikhan Byambasuren, Antonio Montagnoli, Donato Chiatante, Batkhuu Nyam-Osor

**Affiliations:** 1Laboratory of Forest Genetics and Ecophysiology, School of Engineering and Applied Sciences, National University of Mongolia, Ulaanbaatar 14201, Mongolia; seroddamba@num.edu.mn (S.-O.B.); enkhchimeg1120@gmail.com (E.T.); saikhnaa80@gmail.com (O.B.); 2Department of Environment and Forest Resources, College of Agriculture and Life Science, Chungnam National University, Daejeon 34134, Korea; 3Department of Forest Biological Sciences, College of Forestry and Natural Resources, University of the Philippines, Los Baños 4031, Philippines; johernandez2@up.edu.ph; 4Institute of Geography and Geoecology, Mongolian Academy of Sciences, Ulaanbaatar 15170, Mongolia; 5Department of Biotechnology and Life Science, University of Insubria, Via Dunant, 3-21100 Varese, Italy; antonio.montagnoli@uninsubria.it (A.M.); donato.chiatante@uninsubria.it (D.C.)

**Keywords:** arid and semiarid, afforestation, desertification, drought effects, compost fertilizer

## Abstract

Desertification is impeding the implementation of reforestation efforts in Mongolia. Many of these efforts have been unsuccessful due to a lack of technical knowledge on water and nutrient management strategies, limited financial support, and short-lived rainfall events. We investigated the effects of irrigation and fertilization on the morphophysiological traits of *Populus sibirica* Hort. Ex Tausch and *Ulmus pumila* L. and to suggest irrigation and fertilization strategies for reforestation. Different irrigation and fertilizer treatments were applied: no irrigation, 2 L h^−1^, 4 L h^−1^, and 8 L h^−1^ of water; no fertilizer, 2 L h^−1^ + NPK, 4 L h^−1^ + NPK, and 8 L h^−1^ + NPK; and no compost, 2 L h^−1^ + compost, 4 L h^−1^ + compost, and 8 L h^−1^ + compost. The leaf area (LA) and specific leaf area (SLA) of both species responded positively to 4 and 8 L h^−1^. Results also showed that the addition of either NPK or compost to 4 or 8 L h^−1^ irrigation resulted in a higher LA, SLA, and leaf biomass (LB). Total chlorophyll content decreased with irrigation in both species. The same pattern was detected when a higher amount of irrigation was combined with fertilizers. Lastly, we found that both diurnal and seasonal leaf water potential of plants grown in 4 or 8 L h^−1^ were significantly higher than those of plants grown in control plots. Therefore, 4 or 8 L h^−1^ with either NPK or compost has shown to be the optimal irrigation and fertilization strategy for the species in an arid and semiarid region of Mongolia. Results should provide us with a better understanding of tree responses to varying amounts of irrigation with or without fertilizer in pursuit of sustainable forest management in arid and semiarid ecosystems.

## 1. Introduction

Mongolia is a forest-poor arid and semiarid country, with annual precipitation of only 246.10 mm [1]. Desertification in Mongolia has proceeded rapidly and is becoming the major environmental problem impeding the success of reforestation efforts in the county [2,3,4]. Studies have proven that afforestation and reforestation are the most practical approaches for combatting desertification [5,6,7]. Thus, the Mongolian government has initiated the planting of more than 20 million seedlings since the 1980s; however, the reforestation area remains smaller than the deforested area [8,9], and many afforestation efforts have been unsuccessful [10]. The major reasons for failures in afforestation efforts include lack of technical knowledge on water and nutrient management strategies, limited financial support, and short-lived rainfall events or water limitation [9,11]. Therefore, information on appropriate irrigation and fertilizer management strategies could significantly contribute to effective and sustainable afforestation and reforestation initiatives in arid and semiarid areas such as Mongolia.

The most important environmental factors affecting plant growth are moisture and nutrient availability and their interactions in soil. Drought alters the photosynthetic machinery of plants, chlorophyll synthesis, biomass allocation, and other major physiological activities of plants [12,13,14]. The interacting effects of water shortage and nutrient deficiency limit plant growth and forest plantation productivity, particularly in desertified areas [15,16]. An experiment has shown that both fertilizer and irrigation amount explained the increase in chlorophyll content of plants under limited irrigation conditions in China, such that the no fertilizer and irrigation treatments resulted in the lowest chlorophyll content [12]. A study has also found that the medium amount of both fertilizer and water resulted in higher growth performance, number of new shoots, chlorophyll content, and leaf soluble sugar concentration compared with high and low amounts of fertilizer and irrigation treatments [17]. Further, in a study by Basal and Szabo [18], the morphophysiological traits were significantly altered by both the rates of fertilization and irrigation regimes. Some plants have adaptive mechanisms to counteract or respond to fluctuating amounts of nutrients and soil moisture, such as plasticity development, osmoregulation, and cell turgidity even at more negative water potentials [19,20]. However, these adaptive mechanisms are still unknown in many forest tree species, particularly in arid and semiarid areas.

In Mongolia, two of the major challenges in implementing reforestation programs include the high tree-seedling mortality during the planting stage and deterioration of soil fertility or soil nutrient management. The introduction of irrigation and fertilization to reforestation efforts in the region inevitably leads to unwanted ecological consequences, such as water table variations, salinization, and pollution, making the effort unsustainable. Thus, the objective of the study was to investigate the effects of irrigation and fertilization on the growth and morphophysiological traits of *Populus sibirica* Hort. Ex Tausch and *Ulmus pumila* L. and to suggest optimal irrigation and fertilization strategies for using the two species in reforestation. The present study should provide us with accurate irrigation and fertilization recommendations for *P. sibirica* and *U. pumila* in pursuit of sustainable forest management in arid and semiarid regions.

## 2. Results

The interaction of irrigation and fertilization generally had no significant effect on the leaf area (LA) of *P. sibirica* and *U. pumila* (Figure 1; Appendix A). However, the leaf area (LA) varied significantly by the main effects of either watering and fertilizer treatments in both species, particularly in the first growing season (Figure 1; Appendix A). The 2, 4, and 8 L h^−1^ treatments resulted in similar LA in *P. sibirica* and *U. pumila.* The positive effect of NPK application on LA was best observed in 4 and 8 L h^−1^ irrigation treatments, whereas that of compost (COMP) was best observed in 8 L h^−1^, particularly in the second growing season. The CONT (only rainfall) with or without fertilizer resulted in the lowest LA in all species and experimental periods.

Similarly, there was no significant interaction effect of irrigation × fertilization on specific leaf area (SLA) in all species, except in 2019 for *U. pumila* (Figure 2; Appendix A). Either irrigation or fertilization as the main factor significantly affected the SLA of *P. sibirica* and *U. pumila.* Specifically, the 4 and 8 L h^−1^ treatments yielded larger SLA compared with CON in both species. The largest SLA was detected in 8 L h^−1^ + NPK or COMP in *P. sibirica* across experimental periods. In *U. pumila,* 4 and 8 L h^−1^ + NPK generally resulted in a larger SLA compared with CON. The 2 L h^−1^ + COMP consistently yielded larger SLA across experimental periods compared with the other treatments in the case of *U. pumila.*

The effect of irrigation × fertilization on leaf biomass (LB) was significant in the 2019 growing season (Figure 3; Appendix A). Leaf biomass was the highest in 4 and 8 L h^−1^ + NPK or COMP for *P. sibirica* and in 2 L h^−1^ + NPK and 8 L h^−1^ + COMP for *Ulmus pumila* in 2019. In 2018, the main effects of either irrigation or fertilization were significantly different across treatments. All irrigation-alone treatments yielded higher LB compared with CONT in both species. Similarly, all additions of either NPK or COMP resulted in a higher LB than CONT, particularly in 4 and 8 L h^−1^.

We detected significant effects of the interaction of irrigation and fertilization on the total chlorophyll content and chlorophyll a/b ratio in all species (Figure 4 and Figure 5; Appendix A). A high amount of irrigation (8 L h^−1^) with NPK resulted in a significantly lower total chlorophyll content in 2018, but total chlorophyll content increased in 2019 for *P. sibirica.* Contrarily, addition of COMP in 8 L h^−1^ resulted in a significantly higher total chlorophyll content in 2018, but total chlorophyll content decreased in the following year for both *P. sibirica* and *U. pumila*. Noticeably, seedlings grown in 2 L h^−1^ with NPK or COMP generally had a higher total chlorophyll content compared with seedlings grown in other treatments, particularly in 2019 for both species. The chlorophyll a/b ratio of *P. sibirica* and *U. pumila* was generally the highest in irrigation with NPK or COMP, particularly in 4 and 8 L h^−1^ (Figure 5).

Diurnal water potential declined to as low as −1.2 to −1.9 (MPa) in all plots without irrigation, and this was significantly lower (more negative) than those in the irrigation treatments, particularly in 8 L h^−1^ in both experimental periods (Figure 6). In all species, the lowest declines detected in all treatments were at 13:00 and 15:00 in 2018 and 2019, respectively. The seasonal midday (ψ_m_) leaf water potential was consistently the highest at 8 L h^−1^ followed by 4 L h^−1^ from June to August in 2018 for *P. sibirica.* In *U. pumila*, seasonal midday (ψ_m_) leaf water potentials varied significantly by treatments in August; i.e., 4 L h^−1^ had the highest ψ_m_, followed by 8 and 2 L h^−1^, and CONT had the lowest (Table 1). In July 2019, the highest predawn (ψ_p_) and midday (ψ_m_) leaf water potentials of *U. pumila* were observed in 8 and 4 L h^−1^ (Table 2).

## 3. Discussion

### 3.1. Effects of Irrigation and Fertilizer Treatments on Morphological Traits and Growth of P. sibirica and U. pumila

In this study, LA and SLA of both species responded positively to the 4 and 8 L h^−1^ irrigation treatments. Similar studies also reported positive responses of morphological traits to high soil moisture content [21,22]. Here the LA and SLA in the control plots continued to have the lowest values across treatments and years, and this confirms the results of many studies about the effects of drought on leaf traits. Specifically, water stress resulted in the lowest increase in leaf area development [11,23,24], and leaf size tended to decrease with decreasing water supply [25]. A study found that leaf area was strongly negatively correlated with leaf water loss [26], suggesting that bigger leaves of *P. sibirica* and *U. pumila* in 4 and 8 L h^−1^ irrigation levels may be less prone to water loss via evapotranspiration.

The efficient use of irrigation and fertilizer has recently gained much attention in reforestation efforts in arid regions because of the highly variable or limited rainfall events [11]. In this study, we revealed that the interaction of 4 and 8 L h^−1^ with NPK or COMP positively increased LB of all species. This can be attributed to enhanced cell division and physiological activities under water-sufficient or high turgor pressure conditions and improved nutrient availability [27,28]. Our result is consistent with that of some water and fertilizer experiments conducted in drylands [27,29,30]. Similarly, Zhang and Xi [31] observed that the aboveground plant biomass increased when precipitation increased and decreased when precipitation fluctuated significantly. A significant increase in aboveground biomass and leaf area was also detected when the plants were supplied with nitrate, and such a result was associated with the effects of nitrate on cell turgor pressure [32]. Thus, the current result suggests that sustainable reforestation in arid and semiarid regions in Mongolia can be achieved with the use of a combination of the appropriate amount of water and fertilizer. This is because excess water and application of both organic and inorganic fertilizers can exacerbate land degradation and environmental damage in arid and semiarid regions.

### 3.2. Effects of Irrigation and Fertilizer Treatments on Physiological Traits of P. sibirica and U. pumila

Contrary to our expectation, CONT and a low amount of irrigation (2 L h^−1^ with NPK or COMP) have generally shown to have a higher total chlorophyll content compared with a high amount of irrigation with/without NPK or COMP, particularly in 2019 for both species. The result of the present study agrees with the findings of Hassanzadeh et al. [33], who reported that irrigating the plots under flooding conditions led to a decrease in chlorophyll content and senescence, whereas drought stress increased chlorophyll content. Our result is attributable to the effects of moisture on photochemical activity and chlorophyll synthesis in leaves [34,35]. Conditions such as excess water and nutrient availability may have hindered the chlorophyll synthesis of irrigated plants, particularly those grown in 4 or 8 L h^−1^ with NPK and compost. Several studies explained that low levels of chlorophyll in leaves might be poorly related to water conditions in the field because of interacting effects of the other environmental factors, such as soil characteristics, light, and air temperature [36,37]. A change in the amount of far-red radiation and light reflectance of leaves was also cited as one of the reasons for the change in chlorophyll content with increasing soil moisture [38]. A study by Hamblin et al. [39] also mentioned that the reduction in leaf chlorophyll content can reduce the heat load, thereby reducing water requirement to cool leaves.

Nitrogen is an essential part of the chlorophyll molecule; hence, the lack of it in the mineral nutrient supply may significantly affect chlorophyll synthesis in plants. Here we revealed that adding NPK or compost to a higher amount of irrigation significantly decreased the chlorophyll content compared with CONT in all species, implying that some factors such as mineralization, leaching, and volatilization may have come into the picture. Because of the high amount of irrigation, the applied fertilizer may have been leached out, leading to N deficiency in plants. Another possible reason is that the high amount of irrigation may have facilitated the mineralization of the applied fertilizer in the soil, leading to a luxury consumption of N by plants. Such a luxury consumption produced excess N that may not be metabolized into functional or structural compounds necessary for chlorophyll synthesis. One study, however, reported that the stress-induced loss of chlorophyll was not linked to a lack of nitrogen [40]. This is probably because of the effect of water stress on nitrogen mineralization and uptake. Further, the significant reduction in leaf chlorophyll content may be a consequence of increasing the LA, SLA, and LB of the plants grown in a higher amount of irrigation with NPK or compost. A study has shown that reduced leaf chlorophyll content per unit leaf area was associated with high growth performance [39,41].

Lastly, we found that plants grown in 4 or 8 L h^−1^ had significantly higher diurnal leaf water potential than those grown in CONT plots. Seasonal leaf water potentials were also generally higher in 4 and 8 L h^−1^ during summer or growing season in 2018 and 2019. Noticeably, the decline in Ψ in 4 and 8 L h^−1^ plots, particularly during 13:00 to 15:00 time periods, was able to be reversed to the normal condition, suggesting effective recovery of Ψ from diurnal water stress. Our findings were consistent with the results reported in many studies (e.g., [42,43]). Results further imply that both species responded positively to irrigation treatments and negatively to water shortage, indicating strong plasticity in response to water availability fluctuations. The species has shown an ability to cope with stressful events in arid and semiarid ecosystems that are likely to occur annually, particularly during the growing season. Such an ability can be used to optimize irrigation applications depending on the actual plant’s needs and status.

## 4. Materials and Methods

### 4.1. Site Description

The study was conducted in Lun soum, Tuv province, Mongolia (47°52′15.43″ N, 105°10′46.4″ E), which is 135 km west of Ulaanbaatar, with an elevation of 1130 m a.s.l (Figure 7). The study area is characterized by the Middle Khalkha dry steppe, which is a region that has been greatly degraded by intense livestock grazing [44]. The site is dominated by xerophytic and mesoxerophytic graminoids, such as *Stipa sareptana* subsp. *krylovii* Roshev., *Cleistogenes squarrosa* Trin., and *Leymus chinensis* Trin. [45]. Kastanozem type (Loamic), characterized by a lack of profile differentiation in different horizons [46], is the soil type in the study site. The topsoil and subsoil hardnesses are 4.5 and 1.7 kg cm^−2^, respectively [47]. The soil in the study site is generally sandy (69–78% sand, 12–22% silt, and 9–11% clay) with a pH ranging from 7.17–8.05 and organic matter content of 0.248–0.880% [3].

Generally, the growing season in Mongolia starts in May or June and ends officially when the first frost event occurs in September [48]. During the experiment, the mean temperature and the mean precipitation were 16.78 °C and 183–222 mm, respectively. Environmental data were obtained using the HOBO climate data logger installed in the study site (Figure 8). The highest mean temperature (i.e., 21.4 °C) and precipitation (i.e., 93.8 mm) were detected in July and June. The lowest mean temperature (i.e., 9.9 °C) and precipitation (i.e., 0 mm) were detected in May and September, respectively (Figure 8).

### 4.2. Experimental Materials and Design

The study was conducted in two consecutive growing seasons, i.e., May–September 2018 and 2019. We used a total of 512 two-year-old nursery-growth seedlings of *Populus sibirica* and *Ulmus pumila* with initial root collar diameter (RCD) and height of 0.51 ± 0.02 mm and 68 ± 2.94 cm and 0.33 ± 0.01 mm and 51 ± 1.14 cm, respectively. *U. pumila* seedlings were grown from seeds; *P. sibirica* seedlings were obtained from nodal cuttings. The species have an initial root length of 21.50 to 25.9 cm [49]. Thereafter, seedlings were planted into round holes that were 60–70 cm deep and 50–60 cm wide and were irrigated using the same amount of water across treatments during the acclimatization period. After one month, four different irrigation regimes were applied: no irrigation (control or 0 L water h^−1^), 2 L h^−1^ = 0.25 mm m^−2^, 4 L h^−1^ = 0.5 mm m^−2^, 8 L h^−1^ = 1.0 mm m^−2^. A 5 h duration of watering was done twice a week throughout the course of the experiment, using a drip irrigation system connected to a water tank with a capacity of 25 tons. We also used two different types of fertilizers (i.e., 120 and 500 g tree^−1^ of NPK and compost), which were mixed with natural soil and added into the holes before transplanting. The holes were 60–70 cm deep and 50–60 cm wide. NPK consisted of solid granules of the mixture of nitrogen, phosphorus, and potassium following the ratio of 12:16:4. The compost consisted of well-decomposed sheep manure with the following chemical characteristics: 7.4 pH; 18.0–25.0% organic matter content; 5.0–7.0 g kg^−1^ nitrogen content; and total Ca, Mg, K, and Na contents of 9.29, 7.02, 9.18 and 0.05 g kg^−1^, respectively. A total of 12 plots (20 m × 10 m in area) were established for each species in the study site, i.e., 4 plots for irrigation-alone treatments, 4 plots for irrigation + NPK fertilizer treatments, and 4 plots for irrigation + compost treatments. These plots were positioned following a 2.5 m distance between trees and a 2.5 m distance between rows (Figure 7C). We used 32 seedlings for irrigation-alone treatments and 16 trees for irrigation + fertilizer/compost treatments [49,50].

### 4.3. Measurements of Morphophysiological Traits

We collected 1200 healthy and fully expanded leaves (i.e., 4 leaves × 5 replicates × 3 trees × 10 treatments × 2 species) for the determination of leaf area (LA), specific leaf area (SLA), and leaf biomass (LB). We did not collect leaves from controlled plots because all seedlings, if not already dead, were not in good condition at the end of the experiment. Collected samples were first sealed in ziplock bags and stored in a cold container during transport until further analysis in the laboratory. The leaves were photo-scanned using an HP LaserJet scanner (M1132 MFP, Palo Alto, CA, USA) with a 600 dpi resolution. ImageJ processing software was used to analyze the LA of the samples collected following the procedures in Schneider et al. [51] and the modified procedure in Hernandez et al. [52].

In terms of LB and SLA, leaves were first oven-dried at 65 °C for three days and then weighed using a high-precision electronic scale (d = 0.001 g, Discovery Semi-Micro and Analytical Balance, Ohaus Corp., Switzerland). The SLA was calculated using the procedure and equation (i.e., SLA = LA (cm^2^)/LDM (mg)) in Li et al. [53].

Across treatments, we randomly selected 18 seedlings (i.e., 3 tress × 3 replicates × 2 species), from which 1–5 healthy, fully expanded, and sun-exposed leaves were collected for chlorophyll content determination. The samples were first put in an ice-cooled box until actual extraction in the laboratory using the spectrophotometry method. A known amount of leaf tissue (i.e., 0.10 g) was ground and then suspended in 10 mL of 80% acetone, mixed well, kept at 4 °C overnight in the dark, and centrifuged (5000 rpm, 2 min), and the supernatant was obtained [54]. Thereafter, the absorbance was read at 663 and 646 nm in a spectrophotometer (Genesys 10S UV-Vis Thermo Science, Waltham, MA, USA). The total chlorophyll and chlorophyll *a* and *b* concentrations were determined by the following equations [11,55]:Chlorophyll *a* (C_a_, µg/mL) = 12.21 A_663_ − 2.81 A_646_
Chlorophyll *b* (C_b_, µg/mL) = 20.13 A_646_ − 5.03 A_663_
Total chlorophyll (C_a+b_, µg/mL) = C_a_ + C_b_
Chlorophyll *a/b* ratio = C_a_/C_b_

To determine the plant water status, three healthy trees with three replicates from each species and treatment were randomly selected for the determination of leaf water potential (ψ, MPa). In this study, the daily maximum of turgor condition and the daily minimum of turgor condition were assumed to be those at predawn (ψ_p_) and midday (ψ_m_), respectively [11]. Following the procedures in Scholander et al. [56], the ψ_p_ and ψ_m_ leaf water potentials were determined in healthy, fully expanded, and sun-exposed apical leaves using a pressure chamber (Model 1505D EXP, PMS Instrument Company, Albany, OR, USA) (ca. 1.5 m above the ground). Leaves were cut in the petioles and placed inside the chamber, with the cut end protruding from the seal. Measurement was done at the study site.

### 4.4. Statistical Analysis

Two-way analysis of variance (ANOVA) was employed to determine the interacting effects of irrigation and fertilization treatments on the growth and morphophysiological traits measured for each species. To assess multiple comparisons among the treatments, Duncan’s multiple range test (DMRT) was used. All the statistical analyses were run using the Statistical Analysis Software (SAS) package [57].

## 5. Conclusions

In this study, the LA, SLA, and LB were higher in 4 and 8 L h^−1^ treatments with or without NPK and compost when compared with control treatments in all species. However, the total chlorophyll content significantly declined in a higher amount of irrigation with either NPK or compost. Because the survival rate was still high during the course of the experiment, such a decline was seen as a positive response of the species to cope with the changing soil moisture and nutrient conditions. Further, the recovery of leaf water potential of plants grown under 4 or 8 L h^−1^ after a significant decline during 13:00 to 15:00 time periods indicated a certain degree of plasticity of the species in response to fluctuating soil moisture conditions. Therefore, we suggest 4 or 8 L h^−1^ with NPK or compost as the optimal irrigation and fertilization strategy for *P. sibirica* and *U. pumila* in reforestation sites in the arid and semiarid region of Mongolia. The present study should provide us with a better understanding of tree responses to varying amounts of irrigation with or without fertilizer, particularly in pursuit of sustainable forest management in arid and semiarid ecosystems.

## Figures and Tables

**Figure 1 plants-10-02407-f001:**
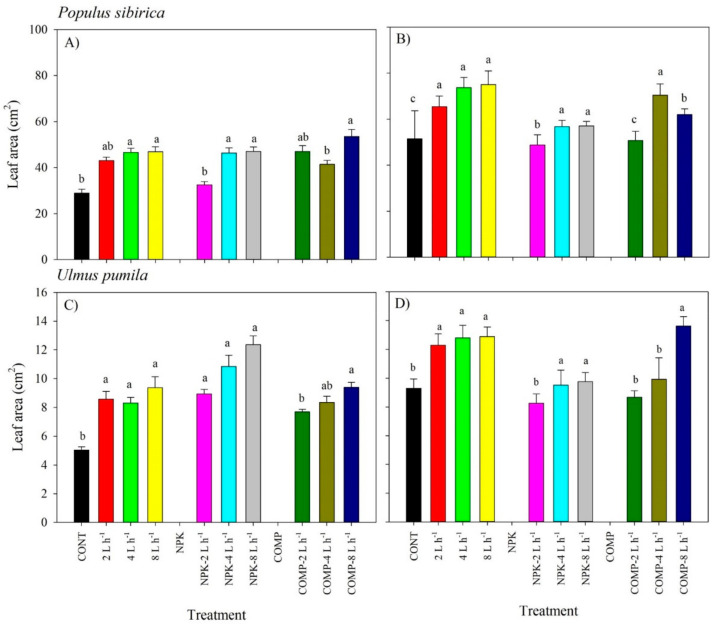
Leaf area (LA) of *Populus sibirica* and *Ulmus pumila* across different irrigation and irrigation + fertilizer treatments (0, 2, 4, 8 L h^−1^; NPK alone, NPK + 2, 4, 8 L h^−1^; COMP alone, COMP + 2, 4, 8 L h^−1^) measured in July 2018 (**A**,**C**) and 2019 (**B**,**D**). Different lowercase letters within each species indicate significant differences across the treatments at α = 0.05. Vertical bars represent standard errors.

**Figure 2 plants-10-02407-f002:**
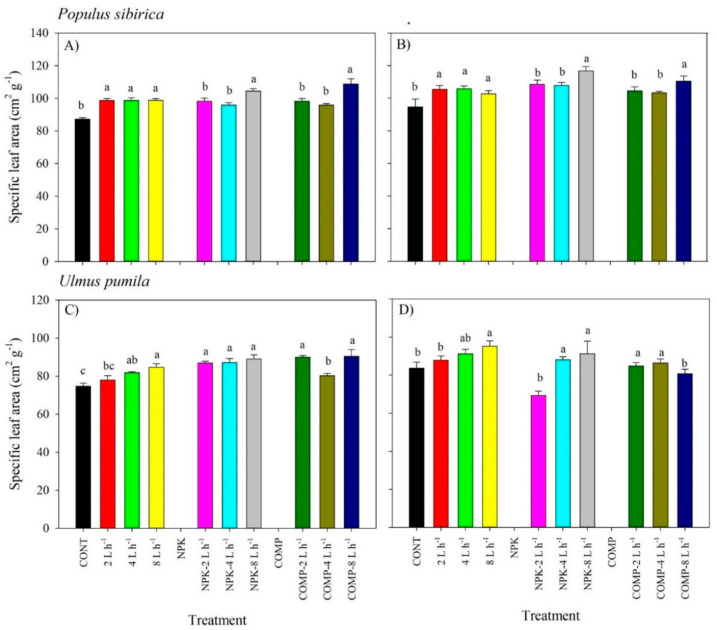
Specific leaf area (SLA) of *Populus sibirica* and *Ulmus pumila* across different irrigation and irrigation + fertilizer treatments (0, 2, 4, 8 L h^−1^; NPK alone, NPK + 2, 4, 8 L h^−1^; COMP alone, COMP + 2, 4, 8 L h^−1^) measured in July 2018 (**A**,**C**) and 2019 (**B**,**D**). Different lowercase letters within each species indicate significant differences across the treatments at α = 0.05. Vertical bars represent standard errors.

**Figure 3 plants-10-02407-f003:**
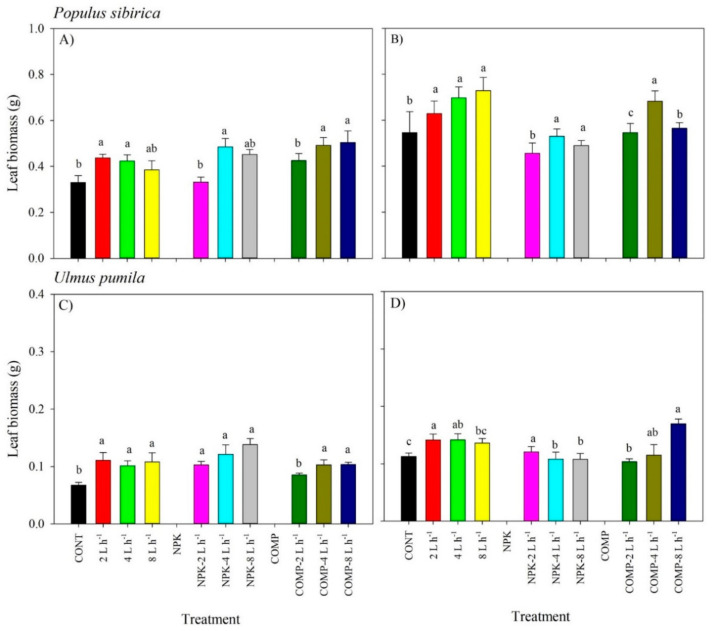
Leaf biomass (LB) of *Populus sibirica* and *Ulmus pumila* across different irrigation and irrigation + fertilizer treatments (0, 2, 4, 8 L h^−1^; NPK alone, NPK + 2, 4, 8 L h^−1^; COMP alone, COMP + 2, 4, 8 L h^−1^) measured in July 2018 (**A**,**C**) and 2019 (**B**,**D**). Different lowercase letters within each species indicate significant differences among the treatments at α = 0.05. Vertical bars represent standard errors.

**Figure 4 plants-10-02407-f004:**
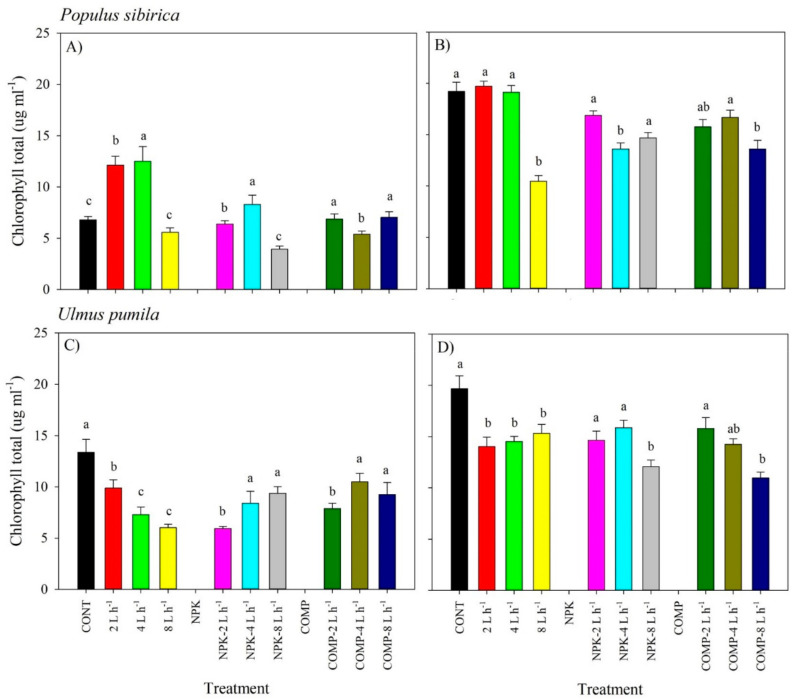
Total chlorophyll content of *Populus sibirica* and *Ulmus pumila* across different irrigation and irrigation + fertilizer treatments (0, 2, 4, 8 L h^−1^; NPK alone, NPK + 2, 4, 8 L h^−1^; COMP alone, COMP + 2, 4, 8 L h^−1^) measured in July 2018 (**A**,**C**) and 2019 (**B**,**D**). Different lowercase letters within each species indicate significant differences across the treatments at α = 0.05. Vertical bars represent standard errors.

**Figure 5 plants-10-02407-f005:**
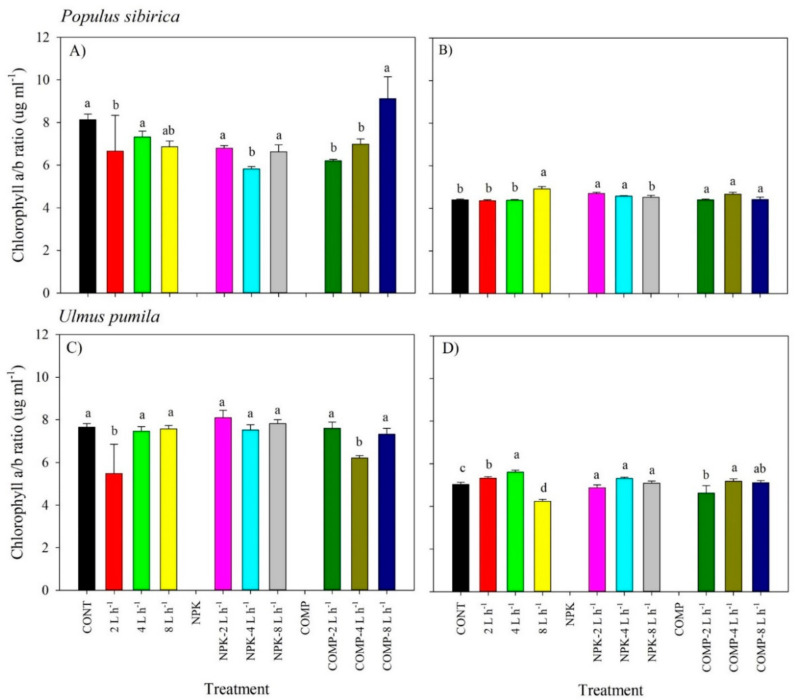
Chlorophyll a/b ratio of *Populus sibirica* and *Ulmus pumila* across different irrigation and irrigation + fertilizer treatments (0, 2, 4, 8 L h^−1^; NPK alone, NPK + 2, 4, 8 L h^−1^; COMP alone, COMP + 2, 4, 8 L h^−1^) measured in July 2018 (**A**,**C**) and 2019 (**B**,**D**). Different lowercase letters within each species indicate significant differences across the treatments at α = 0.05. Vertical bars represent standard errors.

**Figure 6 plants-10-02407-f006:**
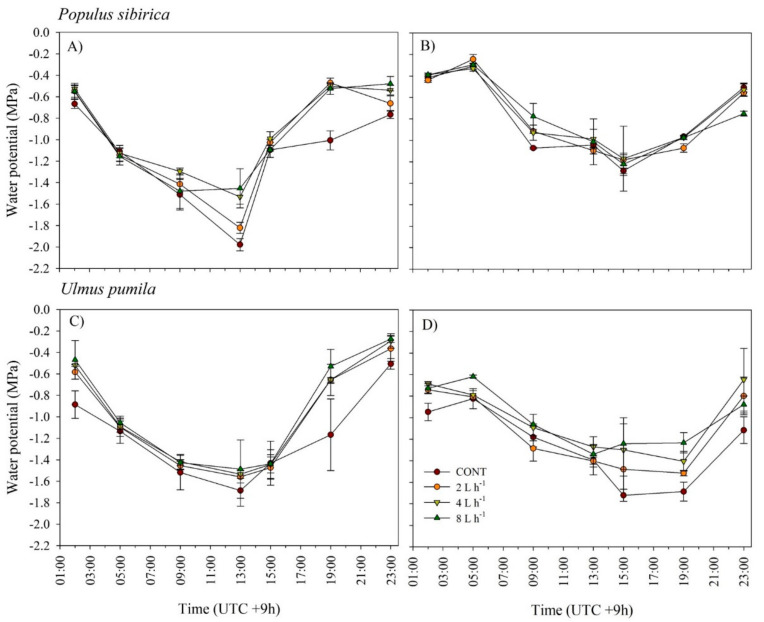
Diurnal variation in leaf water potential (Ψ) of *Populus sibirica* and *Ulmus pumila* across different irrigation treatments (0, 2, 4, 8 L h^−1^) measured in July 2018 (**A**,**C**) and 2019 (**B**,**D**). Vertical bars represent standard errors.

**Figure 7 plants-10-02407-f007:**
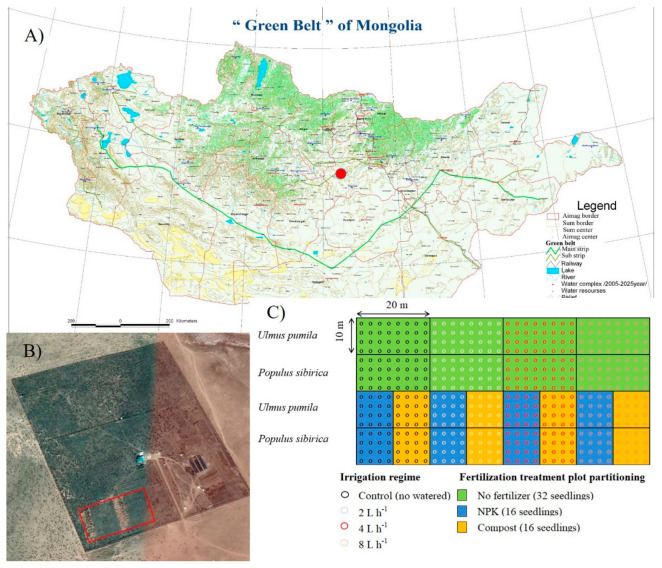
(**A**) Location map of the study site in Mongolia; (**B**) close-up aerial photo of the study area; (**C**) experimental setup.

**Figure 8 plants-10-02407-f008:**
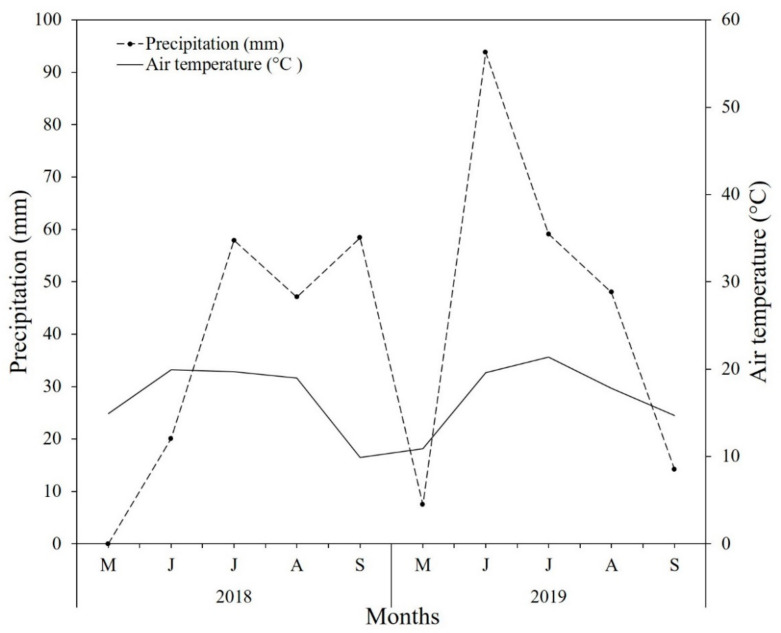
Mean monthly air temperature (solid line) and precipitation (broken, dotted line) in the experimental site during the experiment (i.e., May–September 2018–2019).

**Table 1 plants-10-02407-t001:** Seasonal variation of the predawn (ψ_p_) and midday (ψ_m_) leaf water potentials in *Populus sibirica* and *Ulmus pumila* across different irrigation treatments (0, 2, 4, 8 L h^−1^) measured in 2018. Different lowercase letters indicate significant differences across the treatments at α = 0.05.

Species		Treatment	June	July	August
*U. pumila*	ψ_p_	0 L h^−1^	−1.366 ^a^	−1.133 ^a^	−1.441 ^a^
		2 L h^−1^	−1.096 ^a^	−1.096 ^a^	−1.290 ^a^
		4 L h^−1^	−1.091 ^a^	−1.091 ^a^	−1.279 ^a^
		8 L h^−1^	−1.055 ^a^	−1.055 ^a^	−1.044 ^a^
	Ψ_m_	0 L h^−1^	−1.486 ^a^	−1.686 ^a^	−2.270 ^c^
		2 L h^−1^	−1.334 ^a^	−1.559 ^a^	−1.946 ^b^
		4 L h^−1^	−1.286 ^a^	−1.534 ^a^	−1.526 ^a^
		8 L h^−1^	−1.286 ^a^	−1.478 ^a^	−1.737 ^ab^
*P. sibirica*	ψ_p_	0 L h^−1^	−1.243 ^a^	−1.110 ^a^	−1.441
		2 L h^−1^	−1.143 ^a^	−1.144 ^a^	−1.291
		4 L h^−1^	−1.126 ^a^	−1.126 ^a^	−1.279
		8 L h^−1^	−1.155 ^a^	−1.155 ^a^	−1.044
	Ψ_m_	0 L h^−1^	−1.977 ^c^	−1.977 ^b^	−2.271 ^b^
		2 L h^−1^	−1.821 ^bc^	−1.821 ^b^	−1.946 ^ab^
		4 L h^−1^	−1.533 ^ab^	−1.533 ^ab^	−1.802 ^ab^
		8 L h^−1^	−1.453 ^a^	−1.453 ^a^	−1.711 ^a^

**Table 2 plants-10-02407-t002:** Seasonal variation of the predawn (ψ_p_) and midday (ψ_m_) leaf water potentials in *Populus sibirica* and *Ulmus pumila* across different irrigation treatments (0, 2, 4, 8 L h^−1^) measured in 2019. Different lowercase letters indicate significant differences across the treatments at *α* = 0.05.

Species		Treatment	June	July	August
*U. pumila*	ψ_p_	0 L h^−1^	−0.548 ^a^	−0.823 ^b^	−0.677 ^a^
		2 L h^−1^	−0.520 ^a^	−0.808 ^b^	−0.534 ^a^
		4 L h^−1^	−0.513 ^a^	−0.789 ^b^	−0.516 ^a^
		8 L h^−1^	−0.518 ^a^	−0.619 ^a^	−0.582 ^a^
	Ψ_m_	0 L h^−1^	−1.071 ^a^	−1.394 ^ab^	−1.601 ^a^
		2 L h^−1^	−1.026 ^a^	−1.403 ^b^	−1.570 ^a^
		4 L h^−1^	−1.197 ^a^	−1.272 ^a^	−1.572 ^a^
		8 L h^−1^	−1.145 ^a^	−1.339 ^a^	−1.524 ^a^
*P. sibirica*	ψ_p_	0 L h^−1^	−0.291 ^a^	−0.312 ^a^	−0.503 ^a^
		2 L h^−1^	−0.270 ^a^	−0.245 ^a^	−0.457 ^a^
		4 L h^−1^	−0.192 ^a^	−0.331 ^a^	−0.436 ^a^
		8 L h^−1^	−0.172 ^b^	−0.297 ^a^	−0.575 ^a^
	Ψ_m_	0 L h^−1^	−1.457 ^b^	−1.045 ^ab^	−1.750 ^b^
		2 L h^−1^	−1.369 ^b^	−1.095 ^b^	−1.620 ^ab^
		4 L h^−1^	−1.398 ^b^	−0.988 ^a^	−1.506 ^a^
		8 L h^−1^	−1.022 ^a^	−1.014 ^ab^	−1.550 ^ab^

## Data Availability

The data used are primarily reflected in the article. Other relevant data are available from the authors upon request.

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
