# Peer review of "Effects of Irrigation and Fertilization on the Morphophysiological Traits of Populus sibirica Hort. Ex Tausch and Ulmus pumila L. in the Semiarid Steppe Region of Mongolia"

_plants, 2021, doi:10.3390/plants10112407_

Round 1

Reviewer 1 Report

The manuscript is very well written however the section Materials and Methods has to be transferred under the section Introduction. Introduction is based on recent findings and the description of Materials and Methods is very informative. In addition minor changes are recommended only in Materials and Methods. The results have been appropriately verified by statistical analysis. The presented data are generally well interpreted. Main points are adequately justified and supported by relative references. The conclusion focuses clearly on the main points of the study. This work is recommended for publication

Author Response

Dear the Editor and Reviewers

On behalf of my co-author, I thank the editor and reviewers for the valuable comments on the manuscript entitled “Effects of irrigation and fertilization on the morpho-physiological traits of Populus sibirica Hort. Ex Tausch and Ulmus pumila L. in the semi-arid steppe region of Mongolia”. We have tried to address all the editor’s and reviewers’ concerns in a proper way and believe that the paper has improved considerably.

I would be happy to make further corrections if necessary and look forward to hearing from you soon.

Sincerely,

Batkhuu Nyam-Osor

Professor

National University of Mongolia

Reviewer 2 Report

Not clear on your experimental design. How many replications did you have? The 32 or 16 seedlings are subsamples, not replicates. Results presented are not consistent with the data in Figures.

Line 82: What were the p-values for the main effects and interaction effect?

Line 83: From Figure 1A&B, the 2 L h-1, 4 L h-1, and 8 L h-1 had similar LA.

Lines 86-87: Not consistent with the data presented in Figure 1.

Lines 281-286: What was your experimental design? What was your replication? How many replications did you have for each treatment? From Figure 7C, the 32 or 16 seedlings are subsamples, not replicates.

Line 288: What were your replicates? You have subsamples, but not replicates.

Line 290: What were your control plots? Line 272 you stated “no irrigation (control, 0 L h-1)”, but you collected data for this treatment as shown in Figures.

Line 301: What were your replicates here?

Author Response

(The authors gave the same response as above.)

Reviewer 3 Report

Lines 255-256. This sentence is not clear, please rewrite.

In figure 8, might be interesting report the historical data

How and where the two-year-old nursery-growth seedlings were produced?

Planted manually or mechanically?

Lines 270-271, report the amount of water, please and its chemical characteristics

“irrigation hose system” please specify the system: drip irrigation? Which capacity?

Please add the soil properties (chemical and physical)

Please add when the fertilizers were applied and how

Any plant protection product used?

Any biotic or abiotic stresses recorded?

Please add the solar radiation recorded during the growing season.

In the section M&M, please add how the leaf water potential was measured.

The authors should perform a two-way ANOVA, in fact two effects were investigated (irrigation and fertilization) and it is interesting show their interactions.

Author Response

(The authors gave the same response as above.)

Round 2

Reviewer 2 Report

Still not clear what your experimental design was.